# Multicriteria Model for Determining the Best and Low-Cost Methods of Industrial Heritage Transformation and Utilization under Fuzzy Inputs

**Fanlei Meng \*, Yuxiang Pang and Yeqing Zhi**

School of Architecture and Urban Planning, Beijing University of Civil Engineering and Architecture, Beijing 100044, China

\* Correspondence: mengfanlei@bucea.edu.cn

**Abstract:** The renovation and utilization of industrial heritage are important issues in the field of sustainable urban renewal. The renovation of industrial heritage is influenced by various factors such as the value of a heritage site, its location, the positioning of renovation, and the cost of renovation. Most existing studies focus on the concept of renovation and the establishment of heritage management techniques. However, a low cost in the context of urban sustainability has a greater impact on renovation. Therefore, this paper takes Beijing Xinhua 1949 Cultural and Creative Industrial Park as an example, incorporates the cost–benefit ratio into the plans for a low-cost construction, and proposes a method to evaluate the performance of holistic low-cost construction during the whole life cycle of industrial heritage renovation and utilization. This study uses the AHP method to create an evaluation index system and fuzzy TOPSIS(FTOPSIS) to rank the solutions so as to establish a comprehensive evaluation system to thus evaluate industrial heritage renovation projects that are difficult to fully quantify, with the aim to obtain performance evaluation conclusions. The results of this study suggest that the definition of a low-cost renovation should not be limited to a reduction in investment costs but should also pay equal attention to the cost–benefit ratio before and after renovation, and that functional and spatial sustainability is another feasible strategy for achieving the sustainable renovation of industrial heritage.

**Keywords:** analytic hierarchy process; industrial heritage; life cycle cost; low-cost construction; sustainable urban renewal; fuzzy technique for order preference by similarity to an ideal solution

## 1. Introduction

Industrial heritage is an important stock of spatial resources, and its adaptive use is not limited to only the study of its value at the individual level but extends also to considerations of its potential capacity for sustainable urban regeneration. In recent years, the sustainable reuse and development of industrial heritage have played increasingly important roles in urban renewal in developing countries. For example, millions of square meters of industrial heritage are renovated and utilized every year in China, and issues such as changing the renewal mode, controlling the renovation cost, and sustainably using buildings have become important issues that cannot be ignored in the field of industrial heritage conservation and reuse. It is because of the large number of industrial heritage being renovated and the demand for urban regeneration that previous research and practice have focused on short-term revitalization and utilization, without weighing the cost-benefit of industrial heritage renovation and considering the link between future and urban sustainability, resulting in a failure to meet the requirements of sustainable urban regeneration in the long run.

The low-cost construction mentioned in this article does not only include the construction phase. In the Chinese context, the "construction" of a building generally refers to the entire process, from design and construction to operation and maintenance, and

covers various aspects of the building production process. A low-cost construction strategy during the renovation of industrial heritage not only considers the initial investment costs but also the cost–benefit ratio before and after the renovation, which is an important indicator that measures a low-cost construction; updates the function, space, and even the structure; transforms it into a suitable building model for present and possibly future renovations; facilitates the continuous renewal of the building in the future; and reduces the renovation cost. At this stage, the initial goal of most industrial heritage renovations and renewals is to make a quick profit, which is not in line with the concept of sustainable architecture. Therefore, in the current stage of industrial heritage renovation, there is an urgent need to re-examine the target orientation of renovation and to introduce low-cost construction strategies. A low-cost construction strategy is not limited to a specific process and dimension, is more suitable for the renewal of architectural heritage from a practical point of view, and is more likely to promote the possibility of the future regeneration of industrial heritage.

Based on the existing literature and relevant case studies, this article selects Beijing Xinhua 1949 Cultural and Creative Industrial Park as a case study. This industrial heritage renovation project decided to use a low-cost construction strategy throughout the whole renovation process, beginning at the early stages of renovation, making this case study representative and valuable. From the perspective of a low-cost construction, this study divides the whole life cycle of industrial heritage into three main stages—design, construction, and operation and maintenance—and establishes which performance indicators help maintain a low cost during each stage of the whole life cycle of industrial heritage based on the AHP method. Then, the FTOPSIS method is used to evaluate the different solutions of this case study so as to establish a comprehensive judgment of the proposed low-cost construction system of industrial heritage renovation and utilization. Of course, the establishment of a comprehensive evaluation system is a very important part of this study, but the normal operation of the evaluation system is based on the preliminary theoretical foundations of this study; we combined existing research on industrial heritage and proposed a new perspective on low-cost transformation that serves as the premise behind the need for evaluations, which are an indispensable part of the process and perhaps even more important than even cost calculations. This is one of the ideas that we establish at the beginning of this study and is the basis behind our research logic and the main steps undertaken. The purpose of this study is to investigate the feedback behavior of the low-cost construction strategy during the design, construction, and operation and maintenance phases of the industrial heritage renovation process, and to demonstrate that the cost–benefit ratio is an important concept within the low-cost construction strategy. At the same time, it is concluded that the low-cost construction strategy is more in line with the concept of urban sustainability and more conducive to the long-term revitalization and utilization of industrial heritage than other common renovation strategy models.

## 2. Review of Industrial Heritage Transformation Research

### 2.1. A Survey of Existing Research on Industrial Heritage

Numerous academics domestically and internationally have investigated and used the industrial heritage transformation model and methodology. Most of the works already in existence cover industrial heritage's historical significance, conservation and transformation techniques, and transdisciplinary intersectionality from a broad perspective. Liu Fuying carried out a thorough investigation into the various forms of industrial heritage and created a framework for industrial heritage conservation and a reuse model spectrum based on its scale hierarchy. This framework can assist in transforming and reusing various forms of industrial heritage [1]. Then, Nan Jiang and Jianguo Wang proposed the adaptive reuse of industrial heritage based on value assessment, which helps to avoid the value misjudgment of industrial heritage [2] and thus find means of adaptive reuse relying on value characteristics. Binjun Guan and Fangai Chi propose how industrial heritage can

be integrated into the city or countryside in the right posture, and propose the issue of architectural recycling, which is a topic related to the whole life cycle of industrial heritage.

Oreni, from Politecnico di Milano, used BIM to create an industrial heritage process management model, considerably facilitating the interchange of information between several disciplines [3]. Additionally, the quantitative measurement of industrial heritage rehabilitation and utilization benefits greatly from the use of big data. De Gregorio et al. proposed the combination of industrial heritage and local cultural industry development for adaptive reuse [4], and cultural and creative industrial parks have become the most common renewal strategy in China [5]. How to bring industrial heritage back to life and effectively drive the economy for sustainable urban renewal has become an important topic in the field of industrial heritage today [6,7].

Policy makers and program designers have received methods and ideas for renovation and utilization from the aforementioned studies, but typically because systematic cost considerations are lacking, it is simple to produce high input outcomes and low benefits, which makes it challenging to maintain a good state of sustainable regeneration. In order to finally realize the sustainable transformation and utilization of industrial heritage transformation, it is helpful to study and discuss the low-cost construction of industrial heritage. This helps to find the "optimal solution" in the methods and strategies of industrial heritage conservation and reuse, and to continuously inject life into urban renewal.

### 2.2. Overview of Low-Cost Research in Industrial Heritage Renovation

Researchers have presented the idea of low-cost construction to build low-cost landscapes tailored to sustainable urban regeneration in the domains of landscape, urban parks, etc., according to a search in Web of Science and CNKI using the phrase "low-cost construction" (Figure 1). The reuse value judgment [8] and post-use evaluation [9] of industrial heritage are the current hotspots of domestic scholars' research, and the preliminary literature shows that this research can help further explore the value existence and transformation potential of industrial heritage, but with industrial heritage being an important part of urban stock spatial resources, the key point that cannot be ignored is how to further contribute to industrial heritage through low-cost effective evaluation. The important thing is to effectively examine low-cost renovation solutions for industrial heritage. To accomplish this goal, a thorough study of heritage, using techniques from different disciplines to determine how cost-effective the building behavior was, paves the way for later low-cost development and opens up possibilities for the sustainable use of industrial history [10,11]. The renewal of energy efficiency in buildings has become one of the most important ways to seek cost savings in construction [12,13], combined with new technologies to better control the consumption and production value of costs. The cost of physical construction necessitates sound judgment and situational management throughout the building process, optimizing the structure's optimization, component treatment, and industrial heritage's transformation potential [14]. In general, to accomplish the aim of the low-cost transformation of industrial heritage, it is important to collaborate at all stages and levels, which is also the result of the integration of the economy, technology, culture, and science, but the issue of costs during industrial heritage renovation is now perceived in different ways, and there is a lack of consistent generalization and summary among them.

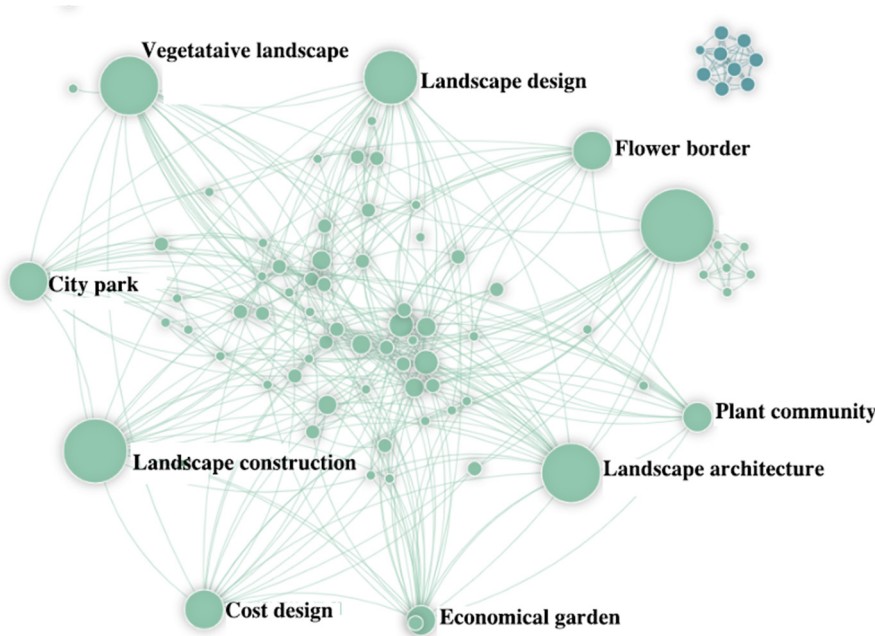

**Figure 1.** Cooccurrence analysis of the subject terms of low-cost construction in Web of Science and CNKI.

### 3. Low-Cost Construction of Industrial Heritage

*3.1. Definition of Industrial Low-Cost Construction*

It is vital to define low-cost construction in the context of industrial heritage renovation in order to fully assess this paper's proposal for affordable industrial heritage renovation and sustainable renewal. In contrast to traditional cost measurement and renovation models, the notion of low-cost construction is defined in this paper as a broad concept of cost-benefit hedging over the whole life cycle of renovation that is flexible, comprehensive, and controllable. We must concentrate on the three key phases of a building's life cycle, namely the design phase, the construction phase, and the operation and maintenance phase, as we investigate the issue of low-cost construction in the process of industrial heritage renovation. We must also further develop a systematic performance evaluation system by examining the relationship between the cost inputs and outputs in various phases. It should be noted that the cost of building demolition should be included in the full life cycle study, however, this paper focuses on how to achieve the appropriate adaptive reuse of industrial heritage and integrate it into urban renewal; therefore, the cost of demolition is outside the scope of this paper.

This study proposes that the low-cost transformation of industrial heritage includes the following two implications:

1. The low-cost construction of industrial heritage renovation should take into account the balance of benefits and costs before and after the renovation as well as the intangible operation and maintenance costs after the renovation, in addition to reflecting the cost status through specific investment amounts, for instance, the development of the abandoned site's spatial vitality and the overall balance between social benefits and cost investment, such as the preservation of historical artifacts and their transformation into cultural property, etc.
2. The costing of engineering costs is different from the performance assessment of the low-cost construction of industrial heritage because it concentrates on the balance of performance in the design phase, construction phase, and operation and maintenance phase of industrial heritage and uses quantifiable data indicators to measure it.

The level and angle of the discussion, as well as an appropriate demarcation criteria, must be clearly defined if we are to analyze and discuss the transformation of industrial heritage from the standpoint of low-cost construction. As a result, this article offers three measurable indicator systems based on the three stages of industrial heritage transformation divided by the whole life cycle cost (LCC): building material and structural trade-off judgments, a comprehensive benefit judgment and balancing, as well as the projected relationships in the design stage, construction stage, and operation and maintenance stage, which are all considered in this judgment. (Figure 2).

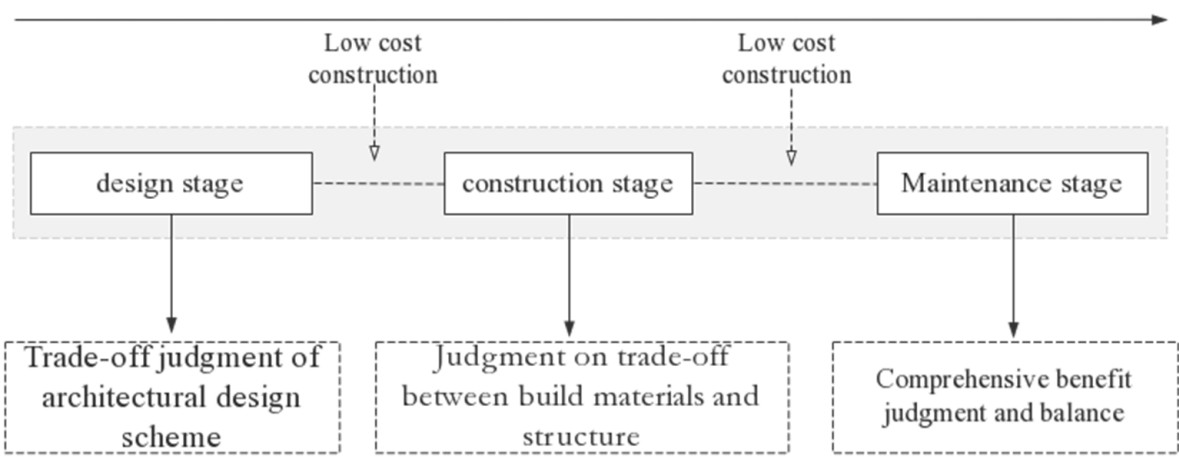

**Figure 2.** Three Stages of Industrial Heritage Transformation.

*3.2. Judgment on the Trade-Offs of Architectural Design Solutions*

When we examined the renovation options for the actual structures, we saw that one of the main causes of the disparate quality of these structures was the complexity and diversity of industrial heritage renovation options. When it comes to renovating industrial heritage buildings, the influx of capital has led to an increase in needless functions and an overburden of work [15]. In actuality, the sustainable renovation of industrial heritage comprises the pre-planning of the building's future variables, the anticipation of future repercussions, and methodical and rational renovation at this point. It also discusses sustainable construction methods and sustainable energy use. This study uses indicators to evaluate the solutions; it does not do so to cast doubt on the reliability of the landed solutions from a scientific standpoint, but rather to evaluate the potential for optimization from the perspective of post-use evaluation and to make inferences from them, as opposed to fending off criticism. This article does not do this type of research since it is impossible to evaluate a building scheme without context. This section focuses on the key indicators that influence the cost input throughout the design phase, with the goal of comparing the options horizontally in the future, looking for strengths that can be learnt from one another and weaknesses that can be avoided.

In the process of industrial heritage renovation, low-cost building tactics and low-maintenance design are both fundamentally indicative of the concept of sustainable architecture. It is a connection in which both parties share the same features. From the original location of the building and the selection of the renovation plan to the prediction of future building operation and maintenance, low-cost construction is a strategic approach presented in connection with sustainable construction in the context of the whole building life cycle, and it is not restricted to cost reduction during the construction phase.

For judging the rationality of architectural design solutions in the design phase, this article establishes the five indicators of building form transformation, the treatment of building materials, the change in building volume ratio, the treatment of structures, and the site's design for comprehensive discussion and application to the calculation of evaluation

indicators below (Table 1). Each indicator makes varied levels of interventions in the cost float of the design phase, impacting the long-term viability of the restored industrial heritage site and emerging as a significant driver of economic growth.

**Table 1.** Discussion of the program trade-off judgment index system.

| Index | Impact on Cost |
| --- | --- |
| Architectural shape transformation | Construction time, material requirements, renovated form factor, and the effect of these on building's energy efficiency, degree of removal of original building structural components, etc. are all influenced by construction difficulty. |
| Treatment methods of building materials | The building's cost and shape will depend on the materials and construction techniques used. There will also be green building materials used. |
| Change of building floor area ratio | The building's floor area to volume ratio has an impact on how the interior of the building changes before and after renovation, and if it is turned into a commercial building, the volume to area ratio has an impact on the structure's ability to generate revenue. |
| The processing method of the structure | Specialized measurements of the physical parameters are needed for the reconstruction of the structure. The architect's suggestion will determine how the original structure will be rebuilt. |
| Site design | Site design is a transformation indication that requires an initial financial investment but will pay off in the long run. |

### 3.3. Building Material and Structural Trade-Off Judgments

The shape chosen by the program has an impact on which materials and structures are a more sensible choice, but other factors such as construction synergy also play a role. Construction materials, technology, equipment, cycle time, staff, and other expenses can all be broken down into the cost consumption during the construction process. The selected materials and the design of the structure form can be successfully implemented in the construction phase of installation, transformation, and erection. The difficulty of the construction will affect the cost of the construction phase of secondary consumption, and the rise in construction difficulty is bound to involve construction technology updates and research and development, so the construction cycle will be lengthened. This also means that energy consumption and carbon dioxide emissions in the construction cycle will once again invariably increase the cost of consumption [16], and the increase in labor costs will be unavoidable due to the elongation of the cycle. Building sustainably may be partially realized by dealing with concerns like the building's energy materials, research, and the application of green building, but owing to the technological barrier, it may be unavoidable that the cost of the construction phase will rise in the near future.

Due to the potential use of green building technology and materials, there are a number of issues that arise during the construction phase that present complex business scheduling and technology application challenges. However, through research, it was discovered that the sustainability achieved through low-cost construction and the sustainability under the green building approach are not mutually exclusive. Rather, their respective foci are different, and the low-cost construction strategy can be shared. When it comes to light pollution, water pollution, and other potential construction issues, as well as cost-management strategies based on green principles, the low-cost construction strategy can be combined with green building construction techniques at various stages [17,18]. A

management system for low-cost construction can be set up for the government and developer firms, and alternative engineering management methods can be used depending on the project's characteristics. Because the return on investment for teams that choose low-cost construction takes some time, the initial cost investment can be supported by specific investments, and the government can also reduce some taxes as a practical measure to assist low-cost renovation.

Overall, the scheduling and coordination of the construction process, which will interact with some of the industry's current technologies, as well as the combining and optimizing of multiple parties, weigh the use of materials and structures in order to achieve the best strategy at the lowest possible cost during this phase.

### 3.4. Comprehensive Benefit Judgment and Balance

Unquestionably, whether for social, environmental, or economic reasons, the revival of industrial heritage helps to revitalize metropolitan regions as a whole. We can use a tool from economics to gauge the project's viability when we talk about the benefits: the benefit–cost ratio [19]. The regeneration of industrial heritage, however, involves more than just monetary purposes; it also ensures the survival of historical culture, urban heritage, and industrial spirit, and has beneficial social implications, among other things. Therefore, the cost consumption and benefits in the operation and maintenance phase should aim for relative balance. This includes the cost of operation, management, and daily maintenance in terms of costs, and the improvement of the area's economic benefits and urban spatial vitality in terms of benefits.

We choose four industrial heritage renovation projects in the central city of Beijing, China as examples, namely Xinhua 1949 Cultural and Creative Industry Park, 1959 INTIMES Creative Industry Park, 77 Cultural and Creative Industrial Park and Beiping Machine Nafu Hutong Store (Figure 3). These four industrial heritage sites were all formerly various kinds of factories, and after being renovated and revived, they have now been reintegrated into the city.

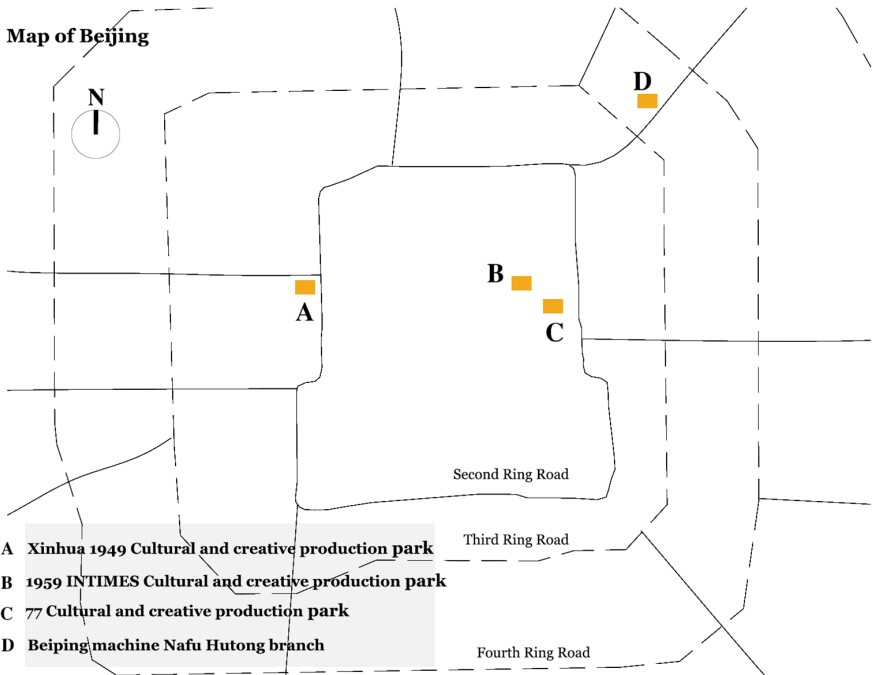

**Figure 3.** Diagram of the location of the four industrial heritage renovation projects.

The economic gain and the preservation of rhetorical memory were the two most significant advantages. The renovation project at Xinhua 1949 Cultural and Creative Industrial Park, for instance, was successful, as evidenced by the high-benefit return that

was attained [20]. The inheritance of historical culture, the preservation of industrial memory, and the improvement of the urban spatial environment were found to be the most obvious benefit returns, and the promotion of urban renewal was not subject to economic feedback, as shown in the following table (Table 2), which compares the four projects.

**Table 2.** Qualitative summary of the visual benefits of four industrial heritage renovation projects.

| Name | General Situation of Transformation | Economic Benefits | Social Benefit | Environmental Benefit |
|---|---|---|---|---|
| A. Xinhua 1949 Cultural and Creative Industry Park | Formerly established in 1949, it is the production workshop and warehouse of Beijing Xinhua Printing Factory. | There are already 66 registered businesses in the park, with a market value of 8–10 billion yuan and an operating income of around 100 million yuan per year. | Protect the industrial remains, keep the industrial memory and historical context. | Creating a good urban environment will attract more investment, which creates a virtuous circle with economic benefits. |
| B.1959 INTIMES Creative Industry Park | The building area is 12,000 m$^2$. Formerly known as China Military Industry No. 125 Factory, Beijing Shuguang Motor Factory of AVIC. | Restaurants, coffee shops, and other common businesses throughout the park are responsible for the park's heat and visible economic benefits. | The original architectural features have been preserved almost entirely, and the older ones are as old as they come. Allow as much architectural memory to be preserved as possible. | There are houses and schools around. It restored the original industrial atmosphere of the building, restrained the later design, and respected the original environmental texture of the city. |
| C.77 Cultural and Creative Industrial Park | The structure is 13,000 m$^2$, once known as the 59-year-old Beijing Offset Printing Factory's former location. Currently, 120 businesses with a total investment of 12 billion yuan have been launched. | The company is the market leader in drama, film, television, and design. The daily rent is 6 yuan/ m$^2$/day. | The park is "small but refined" with a grounded design that allows it to accommodate the lives of the surrounding citizens. | The historic industrial structures were repaired and a significant portion of the urban landscape was restored. |
| D.Beiping Machine Nafu Hutong Store | It was a state-owned copper wire manufacturer in the 1950s, but it is now a bar. | The bar brand effect has evolved into a direct driver of economic benefits, attracting a large number of consumers. | This brand has become one of the representative brands in Beijing's craft beer industry. | After the renovation, the humanistic atmosphere remained. The ancient architectural setting has been protected and passed on. The environment of the old buildings has been preserved and passed on. |

## 4. Evaluation of Industrial Heritage Renovation Plan Based on AHP-FTOPSIS Method

### 4.1. Selection of Research Case

The renovation of the Xinhua 1949 Cultural and Creative Industrial Park, in which the authors had taken part, was chosen as a research case to explore the low-cost evaluation system within the three phases out of the four examples mentioned in the previous article. Because Xinhua 1949 Cultural and Creative Industrial Park is the first officially recognized cultural and creative industrial park project in Beijing and the first low-cost industrial heritage renovation project investigated by enterprises in self-care development, this case

was chosen as an example for in-depth study in the article. After a conversation between the architects and investors early in the project, it was decided that the concept of low-cost renovation would be continued through the entire process of planning, design, building, operation, and maintenance, with the cost–benefit ratio being an important aspect.

The Xinhua 1949 Cultural and Creative Industrial Park is the successor to the 1949-built Xinhua Printing Factory, which had an illustrious history spanning more than 70 years. After being changed into a cultural and creative industrial park, the project was and has been effectively administered and maintained in Beijing's core city for nearly five years. This paper's research team was involved in the project's research, planning and design, proposal selection, and operation and maintenance evaluation stages. There were considerable distinctions between the three choices throughout the project's first selection phase (Figure 4) (Table 3).

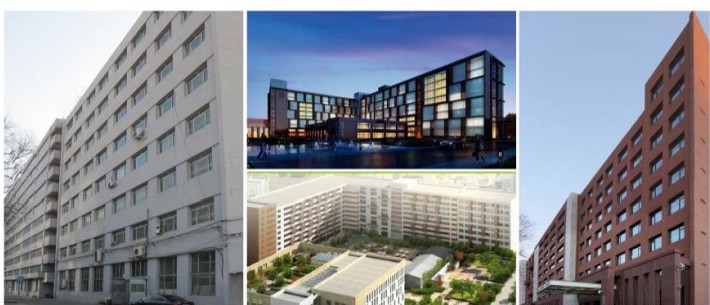

**Figure 4.** The picture on the left shows the original state of the building, the upper middle shows scheme F1, the lower middle shows scheme F2, and the right shows the final implementation, scheme F3.

**Table 3.** Description of the three program profiles.

| Programs | Program Description |
|---|---|
| F1 | The influence of color and shape are overly stressed in the first draft of F1's design. The preservation of the industrial heritage site's originality and the calculation of the renovation cost were not given systematic consideration. |
| F2 | F2 is an updated strategy that includes steps to protect the past and integrate the old and new. The building's structure and materials both received careful consideration. The economic climate was considered. |
| F3 | When considering the viability of the renovated industrial heritage site and the subsequent operating and maintenance costs, the final implemented solution, F3, is more practical in terms of functional use. Additionally, planning for building is simpler. |

### 4.2. Selection of Multi-Criteria Decision-Making Method

Researchers have discovered that multi-criteria decision-making (MCDM), the most popular analysis tool, is more appropriate for use in the field of "sustainable" for measurement and analysis. Given the variety of widely used MCDM methods, selecting a method has evolved into a multi-criteria challenge for various research goals [21] because different methods have different levels of applicability, and it is impossible to directly compare the absolute advantages and disadvantages of the methods. Scholars frequently use a combination of different methods as a metric [22].

The judgment matrix of AHP is simpler to build than the more intricate and systematic ANP method and other methods [23], and the calculation of the system model is more concise. This method can process data qualitatively or quantitatively, but the influence of subjective factors may increase accordingly. However, the value judgment of industrial heritage itself has subjective factors, so AHP has better adaptability to the initial index establishment and weight calculation in the process of industrial heritage transformation [24,25]. The evaluation approach follows the VIKOR principle, which first compromises the limited decision-making strategies by maximizing group utility and lowering individual regret

value [26]. According to the TOPSIS principle, the examined schemes are ordered by how far they are from the positive and negative ideal solutions, with the plan that is the most similar to the positive ideal solution being the best scheme [27], but the evaluation of experts needs a fixed evaluation. As a result, the complete evaluation system developed in this work using the FTOPSIS approach and the AHP method provides the following benefits: (1) When compared to the ANP method, AHP attempts to simplify the problem as much as possible. This makes it appropriate for index judgment and preliminary prediction at the early stages of industrial heritage transformation. (2) FTOPSIS is appropriate for evaluating uncertain problems with diverse standards using quantitative calculation, including uncertain situations like industrial heritage transformation. However, there are also some shortcomings. Less AHP quantitative data is used to build the various index weights, which will affect how rigorous the weights are, and FTOPSIS calculation is more difficult. Generally speaking, the major criterion for method selection is based on how adaptable the approach is to the project being measured.

*4.3. Establishment of Evaluation Index System*

To evaluate the renovation program initially, we must provide various indicators of the cost of the renovation process. We must then assess the program's benefits and drawbacks by computing the weights of the indicator system, which can objectively reflect the variations of the cost-oriented renovation program. The hierarchical analysis method is the foundation of the indexing system. The weights, which can initially ascertain the impact of various sorts of expenses on the end results at various phases, will reflect the percentage of each indicator's influence on the outcomes.

The development of an indicator system has a direct impact on the accuracy of evaluating low-cost construction results in the process of industrial heritage renovation. With the three levels outlined in the preceding part, combined with the indicator system developed before and after the industrial heritage renovation, the evaluation indicator system is explicitly separated into the following three levels (Table 4).

**Table 4.** Evaluation index system of industrial heritage renovation from the perspective of low-cost construction.

| Target Layer | Criterion Layer | Indicator Layer |
|---|---|---|
| A low-cost transformation of the whole life cycle of industrial heritage | B1–design stage (judgment on the trade-offs of architectural design solutions) | C1: Architectural shape transformation<br>C2: Treatment methods of building materials<br>C3: Change of building floor area ratio<br>C4: The processing method of the structure<br>C5: Site design |
| | B2–Construction stage (building material and structural trade-off judgments) | C6: Construction material cost<br>C7: Construction personnel cost<br>C8: Construction Equipment Costs<br>C9: Construction Cycle Costs<br>C10: Construction Technology Costs |
| | B3–Maintenance stage (comprehensive benefit judgment and balance) | C11: Running Costs<br>C12: Management Costs<br>C13: Daily maintenance costs<br>C14: Boosting the economy of the area<br>C15: Enhancing the vitality of urban space |

1. Target layer: evaluation system of whole-life cycle low-cost transformation of industrial heritage (A).
2. Criterion layer: design phase B1, construction phase B2, operation and maintenance phase B3.
3. Indicator layer: Design stage indicators include building form transformation (C1), treatment of building materials (C2), change of building volume ratio (C3), treatment

of building structure (C4), site design (C5). Construction stage indicators: construction material cost (C6), construction personnel cost (C7), construction equipment cost (C8), construction cycle cost (C9), construction technology cost (C10). Operation and maintenance stage: operation cost (C12), management cost (C13), daily maintenance cost (C14), enhance area economy (C15), enhance urban space vitality C16.

*4.4. Evaluation Method of Industrial Heritage Transformation Scheme*

The evaluation content of industrial heritage renovation involves three major types of influence factor sets, Bi (i = 1, 2, 3), in the design phase, construction phase, and operation and maintenance phase; each type of influence factor has its own influence index, C (i = 1, 2, 3, . . . 0.15). They affect each other and there is a fuzzy phenomenon, so this article uses the FTOPSIS method to evaluate these impact indicators and then rank the posting progress of each program from the optimal low-cost transformation.

4.4.1. Establishment of Indicator Weights

The weights of the indicators are determined using AHP, which may systematize complex cost issues to the greatest extent possible and convert challenging-to-quantify multi-objective and multi-criteria decision problems into multi-level, single-objective issues. The project's analysis combines quantitative and qualitative analysis from the core of the evaluation problem with a minimal amount of data.

First, we must construct the judgment matrix according to the one to nine scale method, as shown in Table 5:

**Table 5.** One to nine Scaling Method Extremely Meaningful.

| Scale | Definition | Scale | Definition |
|-------|-----------|-------|-----------|
| 1 | Indicates that two factors, i, j, have the same importance compared to each other. | 9 | Denotes the extreme importance of the former over the latter compared to the two factors, i, j. |
| 3 | Indicates that the former is slightly more important than the latter when compared to the two factors, i, j. | 2, 4, 6, 8 | Denotes the middle value of the above adjacent judgments. |
| 5 | Indicates that the former is significantly more important than the latter when compared to the two factors, i, j | Reciprocal | Opposite comparison of two elements. |
| 7 | Indicates that the former is strongly more important than the latter when compared to the two factors i, j. | - | - |

The next step is to allow subject-matter experts to choose the Brother indicators' importance and create a judgment matrix of evaluation indicators, as represented in the following equation:

$$\mathbf{A} = \begin{bmatrix} b_{11} & b_{12} & \cdots & b_{1n} \\ b_{21} & b_{22} & \cdots & b_{2n} \\ \vdots & \vdots & \ddots & \vdots \\ b_{n1} & b_{n2} & \cdots & b_{nn} \end{bmatrix}, \tag{1}$$

where $\left(b_{ij}\right)_{n \times n}$ denotes the impact scalar value of indicator i on indicator j. Then, the hierarchical single ranking and consistency test are performed for each indicator, and finally, the hierarchical total ranking is performed.

For the results of the expert scoring of the criterion layer (Table 6), $B_{ij}$ denotes the impact scale value of indicator i on indicator j, and $M_f$ denotes the expert. The scoring results were averaged to construct the judgment matrix: $\begin{bmatrix} 1 & 2/5 & 1/2 \\ 5/2 & 1 & 2 \\ 2 & 1/2 & 1 \end{bmatrix}$. After normalizing the data, the maximum characteristic root was $\lambda_{max}$ = 3.025, the consistency index was CI = $\frac{\lambda_{max}-n}{n-1}$ = 0.012, and the consistency ratio was CR = CI/RI = 0.023 < 0.1. Therefore, the judgment matrix and the weights of the target and criterion layers (A-B) were obtained (Table 7). Similarly, the judgment matrix and the weights of indicators in the design stage (Table 8), the judgment matrix and weights in the construction stage (Table 9), and the judgment matrix and weights of indicators in the operation and maintenance stage (Table 10) were all obtained. The comprehensive weights of the industrial heritage renovation evaluation indicators from the perspective of low-cost construction can be obtained from the above analysis (Table 11).

**Table 6.** Criterion layer expert scoring data.

| $M_f$ | $B_{21}$ | $B_{31}$ | $B_{23}$ |
|---|---|---|---|
| $M_1$ | 2 | 1 | 3 |
| $M_2$ | 3 | 1 | 2 |
| $M_3$ | 2 | 2 | 1 |
| $M_4$ | 1 | 1 | 1 |
| $M_5$ | 1 | 3 | 3 |
| $M_6$ | 4 | 5 | 2 |
| $M_7$ | 3 | 1 | 1 |
| $M_8$ | 2 | 2 | 3 |
| $M_9$ | 3 | 3 | 2 |
| $M_{10}$ | 4 | 1 | 2 |

**Table 7.** Criterion layer data.

| B | B1 | B2 | B3 | W |
|---|---|---|---|---|
| B1 | 1 | 2/5 | 1/2 | 0.178 |
| B2 | 5/2 | 1 | 2 | 0.519 |
| B3 | 2 | 1/2 | 1 | 0.304 |

Note: We can find CR = CI/RI = 0.023 < 0.1, which is consistent with the consistency test.

**Table 8.** Design stage data.

| B1 | C1 | C2 | C3 | C4 | C5 | W |
|---|---|---|---|---|---|---|
| C1 | 1 | 1/4 | 1/7 | 1/4 | 3 | 0.067 |
| C2 | 4 | 1 | 1/4 | 1 | 6 | 0.188 |
| C3 | 7 | 4 | 1 | 6 | 7 | 0.540 |
| C4 | 4 | 1 | 1/6 | 1 | 6 | 0.173 |
| C5 | 1/3 | 1/6 | 1/7 | 1/6 | 1 | 0.035 |

Note: We can find CR = CI/RI = 0.0824 < 0.1, which is consistent with the consistency test.

**Table 9.** Construction stage data.

| B2 | C6 | C7 | C8 | C9 | C10 | W |
|---|---|---|---|---|---|---|
| C6 | 1 | 5 | 5 | 3 | 2 | 0.439 |
| C7 | 1/5 | 1 | 1 | 1/3 | 1/5 | 0.082 |
| C8 | 1/5 | 1 | 1 | 1/3 | 1/5 | 0.082 |
| C9 | 1/3 | 3 | 3 | 1 | 3 | 0.250 |
| C10 | 1/2 | 2 | 2 | 1/3 | 1 | 0.148 |

Note: We can find CR = CI/RI = 0.0377 < 0.1, which is consistent with the consistency test.

**Table 10.** Maintenance stage data.

| B3 | C11 | C12 | C13 | C14 | C15 | W |
|---|---|---|---|---|---|---|
| C11 | 1 | 3 | 3 | 1/5 | 5 | 0.200 |
| C12 | 1/3 | 1 | 1 | 1/7 | 3 | 0.087 |
| C13 | 1/3 | 1 | 1 | 1 | 3 | 0.087 |
| C14 | 5 | 7 | 7 | 1 | 8 | 0.586 |
| C15 | 1/5 | 1/3 | 1/3 | 1/8 | 1 | 0.040 |

Note: We can find CR = CI/RI = 0.0465 < 0.1, which is consistent with the consistency test.

**Table 11.** Comprehensive weights for the evaluation index system of low-cost construction of industrial heritage renovation.

| | B1 | B2 | B3 | IIW | | B1 | B2 | B3 | IIW |
|---|---|---|---|---|---|---|---|---|---|
| | 0.178 | 0.519 | 0.304 | | | 0.178 | 0.519 | 0.304 | |
| C1 | 0.067 | — | — | 0.012 | C9 | — | 0.250 | — | 0.130 |
| C2 | 0.188 | — | — | 0.033 | C10 | — | 0.148 | — | 0.077 |
| C3 | 0.540 | — | — | 0.096 | C11 | — | — | 0.200 | 0.018 |
| C4 | 0.173 | — | — | 0.030 | C12 | — | — | 0.087 | 0.061 |
| C5 | 0.035 | — | — | 0.006 | C13 | — | — | 0.087 | 0.061 |
| C6 | — | 0.439 | — | 0.228 | C14 | — | — | 0.586 | 0.178 |
| C7 | — | 0.082 | — | 0.043 | C15 | — | — | 0.040 | 0.012 |
| C8 | — | 0.082 | — | 0.043 | — | — | — | — | — |

Note: The weights of B1, B2, and B3 are 0.195, 0.717, and 0.088, respectively; IIW is the combined weight, which indicates the concatenation of the weights of Ci and Bi.

### 4.4.2. Fuzzy TOPSIS Evaluation Method

The TOPSIS approach, which was first put forth by C.L. Hwang and K. Yoon in 1981, rates assessment objects based on how close they are to idealized targets. The weights between the costs indicated by the various indexes can be effectively employed in the assessment process of the program, boosting the scientific quality of the evaluation by using the FTOPSIS method to evaluate the various index weights produced from the AHP method.

For the set of indicators and rubrics required by the FTOPSIS method, the set of indicators is established as $C = \{C_1, C_2, C_3, \ldots C_{14}, C_{15}\}$, and assuming that the rubrics of the fifteen indicators are good, better, medium, worse, and poor, the set of indicator rubrics should be $V = \{V_1, V_2, V_3, V_4, V_5\}$ = {good, better, medium, worse, poor}, and the corresponding scoring criteria should be $\{V_1 \geq 90, 80 \leq V_2 < 90, 70 \leq V_3 \leq 80, 60 \leq V_4 < 70, V_5 < 60\}$.

1.  Experts are then invited to score the indicators and construct a fuzzy matrix as follows:

$$\mathbf{R} = \left(\mathbf{r_{ij}}\right)_{\mathbf{m \times n}} \begin{bmatrix} \mathbf{r_{11}} & \mathbf{r_{12}} & \cdots & \mathbf{r_{1n}} \\ \mathbf{r_{21}} & \mathbf{r_{22}} & \cdots & \mathbf{r_{2n}} \\ \vdots & \vdots & \ddots & \vdots \\ \mathbf{r_{n1}} & \mathbf{r_{n2}} & \cdots & \mathbf{r_{nn}} \end{bmatrix} \quad (2)$$

where $r_{ij}$ indicates the affiliation of the indicator, $C_{ij}$, to the rubric, $V_j$.

2.  Data values for Equation (2) are transferred by discretely normalizing the data and applying linear variation. The easiest technique to remove the impacts of the magnitude and of the range of data values is by discrete normalization, which keeps the relationships present in the original data. The conversion equation is as follows:

$$\mathbf{Z_{ij}} = \frac{\mathbf{r_{ij} - min}}{\mathbf{max - min}}(\mathbf{i = 1, 2, \ldots, m; \ j = 1, 2, \ldots, n}) \quad (3)$$

3.  Constructing the weight normalization fuzzy matrix is performed as follows:

$$V_{ij} = W_j Z_{ij} \, (i = 1, 2, \ldots, m; \; j = 1, 2, \ldots, n) \tag{4}$$

where $W_j$ represents the combined weight value of the $j_{th}$ indicator, calculated by the AHP-based method above.

4. Determining the positive and negative ideal solutions is performed as follows:

$$P_j^+ = \max\{V_{1j}, V_{2j}, \ldots, V_{nj}\} \; (j = 1, 2, \ldots, n) \tag{5}$$

$$P_j^- = \min\{V_{1j}, V_{1j}, \ldots, V_{nj}\} \; (j = 1, 2, \ldots, n) \tag{6}$$

5. Calculating the distance to the positive (negative) solution for each scenario is performed as follows:

$$d_i^+ = \sqrt{\sum_{j=1}^{n} (V_{ij} - P_j^+)^2} \, (i = 1, 2, \ldots, m; \; j = 1, 2, \ldots, n) \tag{7}$$

$$d_i^- = \sqrt{\sum_{j=1}^{n} (V_{ij} - P_j^-)^2} \, (i = 1, 2, \ldots, m; \; j = 1, 2, \ldots, n) \tag{8}$$

6. Calculating the posting schedule of the ideal solution is performed as follows:

$$C_i = \frac{d_i^-}{d_i^+ + d_i^-} \; (i = 1, 2, \ldots, n) \tag{9}$$

For the selected cases, the impact indicators, and the evaluation set given by the study to score the three programs, the results of the scoring are shown in Table 12.

**Table 12.** Expert scoring results.

| P | C1 | C2 | C3 | C4 | C5 | C6 | C7 | C8 | C9 | C10 | C11 | C12 | C13 | C14 | C15 |
|---|----|----|----|----|----|----|----|----|----|-----|-----|-----|-----|-----|-----|
| F1 | 96 | 85 | 86 | 87 | 94 | 93 | 84 | 92 | 96 | 90 | 81 | 83 | 91 | 87 | 96 |
| F2 | 85 | 90 | 95 | 94 | 89 | 80 | 70 | 89 | 84 | 80 | 94 | 78 | 85 | 96 | 85 |
| F3 | 87 | 93 | 98 | 96 | 87 | 87 | 72 | 84 | 80 | 82 | 83 | 75 | 83 | 94 | 87 |

The findings were then displayed in Table 13 after the scoring results were normalized using Formula (3). The findings following normalization were reported in Table 14 after the normalized data was further adjusted in accordance with Formula (4). For comprehensive Tables 11–13, after calculation, we obtained the progress of the transformation evaluation indicators of schemes F1, F2, and F3 from the optimal target (Table 15) and the ranking of the three schemes.

**Table 13.** Normalized scoring results.

| P | C1 | C2 | C3 | C4 | C5 | C6 | C7 | C8 | C9 | C10 | C11 | C12 | C13 | C14 | C15 |
|---|----|----|----|----|----|----|----|----|----|-----|-----|-----|-----|-----|-----|
| F1 | 0.3582 | 0.3172 | 0.3082 | 0.3141 | 0.3481 | 0.3705 | 0.3717 | 0.3692 | 0.3472 | 0.3571 | 0.3140 | 0.3517 | 0.3514 | 0.3141 | 0.3048 |
| F2 | 0.3172 | 0.3358 | 0.3405 | 0.3394 | 0.3296 | 0.3187 | 0.3097 | 0.3231 | 0.3358 | 0.3175 | 0.3643 | 0.3305 | 0.3282 | 0.3466 | 0.3494 |
| F3 | 0.3246 | 0.3470 | 0.3513 | 0.3466 | 0.3222 | 0.3108 | 0.3186 | 0.3077 | 0.3170 | 0.3254 | 0.3217 | 0.3178 | 0.3205 | 0.3394 | 0.3457 |

**Table 14.** Scoring results after normalization of weights.

| P | C1 | C2 | C3 | C4 | C5 | C6 | C7 | C8 | C9 | C10 | C11 | C12 | C13 | C14 | C15 |
|---|----|----|----|----|----|----|----|----|----|-----|-----|-----|-----|-----|-----|
| F1 | 0.0043 | 0.0105 | 0.0296 | 0.0094 | 0.0021 | 0.0845 | 0.0160 | 0.0159 | 0.0451 | 0.0275 | 0.0057 | 0.0215 | 0.0214 | 0.0559 | 0.0037 |
| F2 | 0.0038 | 0.0111 | 0.0327 | 0.0102 | 0.0020 | 0.0727 | 0.0133 | 0.0139 | 0.0437 | 0.0244 | 0.0066 | 0.0202 | 0.0200 | 0.0617 | 0.0042 |
| F3 | 0.0039 | 0.0115 | 0.0337 | 0.0104 | 0.0019 | 0.0709 | 0.0137 | 0.0132 | 0.0412 | 0.0251 | 0.0058 | 0.0194 | 0.0195 | 0.0604 | 0.0041 |

**Table 15.** Progress and ranking of F1, F2, and F3 indicators from the ideal solution.

| P | $(d_i^+)$ | $(d_i^-)$ | $(C_i)$ | Ranking |
|----|-----------|-----------|---------|---------|
| F1 | 2.78129553 | 1.56811484 | 0.36053504 | 3 |
| F2 | 1.64645963 | 2.22965375 | 0.57522924 | 2 |
| F3 | 1.28238742 | 2.73519736 | 0.68080638 | 1 |

*4.5. Result Analysis*

The previous computation revealed that the building phase of industrial heritage renovation produces a somewhat bigger influence on the criterion layer (with a weight of 0.519), followed by the design phase (with a weight of 0.178), and then the operation and maintenance phase (with a weight of 0.304). It is well known that the indicator weights for each of the three phases are more evenly distributed for the indicator layer. The volume ratio indicator item has the most weight during the design phase, and it is known from the features of industrial heritage renovation that the volume ratio change before and after the renovation has a direct impact on the building's benefits. It can be concluded that industrial heritage renovation drives the area's economy and increases the cost–benefit ratio in order to achieve renewal sustainability, which is a recognized method by experts. The weight of the item of economic indicators to enhance the area is greatest during the operation and maintenance stage. Based on the FTOPSIS method, the ranking of the three options is $C_3$ (0.6808) > $C_2$ (0.5752) > $C_1$ (0.3605). When focusing on the indicators of the three possibilities, it is clear that neither the pursuit of economic benefits nor the return of economic benefits can make up for the loss of cultural, historical, and living components, proving that the ideal goal of low-cost construction is not achieved. Unbalanced stage decisions will have an impact on the overall reuse of industrial heritage effects. Only when the design phase, construction phase, and operation and maintenance phase are all as cost-effective as possible can the aim be reached in the conservation and reuse of industrial heritage. The research methodology does, however, have several drawbacks, such as the inadequacy of the indicator establishment, which prevents a thorough summary of the industrial heritage transition process. Second, there are some flaws in the rigor of the quantitative calculation due to the inadequate data collecting. For instance, each indication in the project's transformation process should be evaluated using specific data in order to serve as the foundation for expert judgment weights.

**5. Conclusions**

This paper proposes three stages of the low-cost transformation and utilization of industrial heritage under the costs of the whole life cycle—the design stage, construction stage, and operation and maintenance stage—and establishes three quantifiable factors—judgment on the trade-offs of architectural design solutions, the building material and structural trade-off judgments, and a comprehensive benefit judgment and balance—in an effort to establish a performance evaluation system of the proposed three stages.

The cost reduction possibilities of the three core phases of industrial heritage renovation in different construction processes are discussed concurrently using a system of three quantifiable indicators. The operation phase produces corresponding benefits to counterbalance some of the costs in addition to cost control through subjective judgment in the design phase, cost reduction through construction coordination in the construction phase, and cost saving through management optimization in the operation and maintenance phase. However, the study has certain limitations. The low-cost construction being considered at this time does not take special heritage values into account as a significant influence on the transformation index because the priceless value of a small portion of industrial heritage can significantly restrict the transformation and the existing index system cannot be adjusted to such an example of industrial heritage. The function of industrial heritage, for China, transformed into the kind of function that was also within certain bounds, and transforming into residential use is still a subject worth discussing, so the index and the

comprehensive evaluation system could hardly be applied to a very small portion of the low-cost transformation evaluation.

Based on the theory and case studies, it was concluded that: (1) The low-cost transformation and exploitation of industrial heritage requires thorough cost and performance analysis, paying close attention to and weighing the numerous advantages produced by the project's transformation and utilization against the project's input costs. Sustainable urban regeneration will benefit from the adaptive reuse of a vast number of industrial heritage sites through low-cost development. (2) The volumetric ratio, spatial form, and functional flexibility of industrial heritage functions and spaces need to be the primary renovation targets if industrial heritage is to achieve sustainable development through low-cost renovation. (3) Material costs account for the greatest proportion of construction costs, but low-cost construction and expensive green building materials are not mutually exclusive, so the necessary cost spending in the early stage will result in significant benefits later on.

Based on the preliminary findings and limitations of the study, the study also points to the direction for further research to follow in the future: (1) What is the range of the cost–benefit ratio of the low-cost renovation of industrial heritage that maximizes the benefits of low-cost construction? Furthermore, what are the general distribution characteristics? We believe that we can delineate the approximate range, which will help to guide the cost targeting and benefit estimation in the early stage of renovation. (2) When the total cost remains constant, one of the directions for future research to follow will be to determine what proportion of the cost of each stage will maximize the benefit of the cost.

**Author Contributions:** Conceptualization, F.M.; methodology, Y.P. and Y.Z.; investigation, F.M. and Y.P.; resources, F.M.; data curation, Y.P.; writing—original draft preparation, F.M. and, Y.P.; writing—review and editing, Y.P.; project design, F.M. All authors have read and agreed to the published version of the manuscript.

**Funding:** This research was funded by the National Natural Science Foundation, grant number 51808021 and funded by the Beijing Municipal Education Commission, grant number KM202310016014.

**Institutional Review Board Statement:** Not applicable.

**Informed Consent Statement:** Not applicable.

**Data Availability Statement:** Not applicable.

**Conflicts of Interest:** The authors declare no conflict of interest.

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
