# Peer review of "Multicriteria Model for Determining the Best and Low-Cost Methods of Industrial Heritage Transformation and Utilization under Fuzzy Inputs"

_sustainability, doi:10.3390/su15043083_

Round 1

Reviewer 1 Report

Dear Authors,
thank you for your interesting paper. My comments are:
- I recommend expanding the Introduction, it is too short and lacks information,
- line 60 - "The", capital "T", the beginning of the sentence,
- line 61 - "In", capital "I", the beginning of the sentence,
- line 66, "[8-11]." - there is a comma, but shouldn't there be a dot as the end of the sentence?
- Fig. 1 - the text lacks a reference to this Fig. 1, a link to this image must be added to the text, or it must be deleted. Is it necessary? What does it represent?
- Fig. 1 - hard to read text, I recommend making the font bigger,
- Fig. 1, there is written "Web Science" - do you mean "We of Science"? Missing "of",
- line 193 - Fig. 3 (after renumbering),
- line 198, Fig. 4 - Fig. 3 is missing, so this should be Fig. 3 - all other Figs. must be renumbered,
- Fig. 4 (now Fig. 3) - hard to read text, I recommend making the font larger,
- line 202, there is a link to (Table 3) - shouldn't it be a link to Table 2?
- chapter 4. is missing, after chapter 3.4 goes chapter 5. - chapters need to be renumbered,
- in chapter 3.4, 4 selected buildings are presented (Table 2), but then in chapter 4.1.1. (5.1.1) Case Selection, only one building is selected from them - why? Why was the case study done on only one building? It is then necessary in Tab. 2 also list the other three buildings?

Best regards.

Author Response

Dear Reviewer:

Now It's the Lunar New Year in China. First of all, I wish you a happy Chinese New Year.

Thank you very much for your comments on the article and for giving us the opportunity to revise it.

We have revised the article point by point according to your comments, please find the details in the attachment.

Sincerely,

Manuscript authors

Reviewer 2 Report

Dear Authors,

1. As the last note in your abstract, please provide a "take-home" message.

2. Rearrange keywords alphabetically.

3. Please do not use abbreviations in keywords.

4. Previous studies must be explained in the introductory part, including their work, innovation, and limits, to demonstrate the research gaps that will be filled in the current study.

5. Reference for figure 1 needs to be given, and the figure is not included in the text.

6. Figure 2 is not in a good resolution, the text is not readable

7. On page 6 in the text it mentions Table 3 but below is Table 2, I think there is a confusion, please check

8. On several pages of the paper the last line is a subsection/subchapter title, it is not possible that the page ends with a subtitle....

9. Section 4 is missing the paper goes directly to section 5

10. Tables 5, 6, 7, 8 and 9 should perhaps be explained in more detail, and also tables 10, 11 and 12.

11. At the end of the Result Analysis part, the present study's limitations must be added.

12. Mention further research in the conclusion section.

13.  Conclusions would be good to be extended.

14. Perhaps the title should be reworded a bit.

As a summary of the review the paper should be reworded in some places and the tables and figures explained in more detail so that the study would have a logical course, also checked that all figures, tables and references are included in the text.

After some reworking of the paper it can be resubmitted.

Regards,

Author Response

(The authors gave the same response as above.)

Reviewer 3 Report

I feel honored to be able to comment on your paper which is Research on Comprehensive Evaluation System of Sustainable Transportation and Utilization of Industrial Heritage Based on AHP-TOPSIS Measurement from the Perspective of low-cost Construction. This manuscript has an interesting and valuable topic. The subject of the manuscript falls within the scope of the journal. This manuscript represents a good contribution to this area of research. The following minor issues should be addressed to improve the quality of this paper:

1. The abstract needs to be improved. The narrative flow of methodology description should give a better feel of the covered topic. Maybe you can refer to this paper: Guo, J., Xiang, P., Wang, X., & Lee, Y. (2022). Predicting and managing megaproject gray rhino risks with IF-ANP and DEMATEL based on panel data. Expert Systems with Applications, 208, 118243.

2. The Introduction section does not clearly describe the necessity and value of the research. The author should better introduce the innovations of the paper compared with existing research in the Introduction section.

3. The objective of this research is not clearly stated. The authors should describe more clearly the main steps of the investigation in the chapter of introduction.

4. In section 5.1.1. Case Selection, the author should explain why the selected cases are representative. Simply introducing the basic situation of the project is not enough to convince readers that this project can represent all projects of the same type.

5. Authors should explain the reasons for choosing AHP and TOPSIS methods in the appropriate sections. AHP and TOPSIS are very traditional MCDM methods. The authors need to explain why AHP and TOPSIS are able to solve the problem of this study. The authors also need to explain the advantages of AHP and TOPSIS compared to other MCDM methods. Author may refer the following articles:

A multi-criteria decision-making approach to help resource-exhausted areas choose suitable transformation templates—The example of Wansheng in Chongqing, China. Ain Shams Engineering Journal, 13(5), 101709.

Analyzing and Controlling Construction Engineering Project Gray Rhino Risks with Innovative MCDM Methods: Interference Fuzzy Analytical Network Process and Decision-Making Trial and Evaluation Laboratory. Applied Sciences, 12(11), 5693.

6. The conclusions should state in a clear and complex manner the findings of the performed study, based on the results obtained. The findings that originate firstly from this research should be better highlighted. Also, and future research direction may be indicated.

Author Response

Dear Reviewer:

Now It's the Lunar New Year in China. First of all, I wish you a happy Chinese New Year.

Thank you very much for your comments on the article and for giving us the opportunity to revise it.We have carefully considered your comments and then revised them carefully.

We have revised the article point by point according to your comments as well as made relevant explanations and please find the details in the attachment.

Sincerely,

Manuscript authors

Reviewer 4 Report

Article

Research on Comprehensive Evaluation System of Sustainable Transformation and Utilization of Industrial Heritage Based on AHP-TOPSIS Measurement from the Perspective of Low-cost Construction

Fanlei Meng, Yuxiang Pang

It is difficult to say whether the article meets the requirements of the journal, but the authors can refine and rewrite it so that it could be published in the future.

The topic touched will be of interest to readers, but its implementation by the authors seems unsuccessful.

The whole work can be divided into two parts: (1) not directly related to the article; (2) directly related to the study.

Consider first the first part, which, as we have already said, can be deleted as useless. It occupies 6 pages (sections 1-4).

The abstract should be rewritten and reflect the main stages of the study. "Comprehensive valuation system," "sustainable conversion and use," "AHP," "TOPSIS," "low-cost construction" are the key words of the theme. However, sections Introduction, Review of studies (industrial heritage, low cost retrofit of industrial heritage) are incomplete, and their content is not relevant to the topic: the assessment of industrial heritage based on the AHP-TOPSIS measurement, complicated by the problem of how to value low cost construction.

Section “3.1. Definition of industrial low-cost construction" is not directly related to the topic, since the authors would have to consistently define "comprehensive pricing system," "sustainable conversion and use," "industrial development," "AHP," "TOPSIS," and only last turn to the problem of assessing low-cost construction. The authors' reasoning is not logical and misleads readers about the subject and object of the study. Figure 2 “Three Levels of Low-Cost Adaptive Use Study of Industrial Heritage” does not directly address the main topic.

Section “3.2. Judgment on the reasonableness of an architectural design scheme” and Table 1 “Discussion of the program rationalization evaluation index system” deviate from the topic, since any evaluation is created on the basis of reasonableness. An unreasonable assessment is not competent and does not require scientific consideration.

Sections “3.3. Discussion of the program rationalization evaluation system,» «3.4. Comprehensive Benefit Assessment and Balance” does not present any credible discussion of the assessment or estimated indicators, determining the relative value of construction depending on costs, various natural and economic conditions and the location of land. We would expect to present an assessment (ideally a comprehensive assessment) of construction, for example, in points and understand what requirements can be placed on firms in terms of costs, benefits, profitability, taxation, etc. However, there is nothing of the kind in these sections. The authors do not fully understand the problem of low-cost construction in China. "Inexpensive" means not related to "green" building and materials throughout their entire life cycle. "Inexpensive" means unsustainable. If there is "unsustainable", then this article cannot be published in this journal.

Table 2 "Benefit Analysis of Four Industrial Heritage Renovation Projects" does not provide any insight into a comprehensive assessment based on the AHP-TOPSIS measurement. We emphasize: no evaluation based on measurement, etc. based on the AHP-TOPSIS and + measurement in terms of low cost building (the reader is left to guess what low cost building in China means… perhaps something that does not fit into the concept of sustainable building due to the cost nature of green building).

The second part

The second part is the part with which, in the proper sense, on can start the review. It includes 5 incomplete pages. In addition, in it we find 10 tables and 1 figure. Word count is 560 or 4200 characters. In our opinion, the article does not meet the requirements of the journal due to its small volume.

The second part begins with section “5.0. Evaluation of an industrial heritage renovation plan based on the AHP-TOPSIS method.”

The author's rating system consists of a total of 15 levels of indices at 3 indicative levels: namely, the design stage (assessment of the rationality of the development scheme) (B1), the construction stage (assessment of the rationality of building materials and structures) (B2) and the operation and services (assessment and balance of integrated benefits) (B3) (Table 4).

However, below we learn that a survey and peer review were carried out. Because of this, we are forced to turn to the analysis of the research methodology.

It must be admitted that the article does not describe it, so readers cannot fully understand how the study was conducted, what stages of analysis were required to study such topics. The procedure for questioning and conducting expert assessments requires certification and publication of the main documents (calculated sample data, questionnaires, results of expert assessments, linguistic assessments of each expert for each criterion of weights of criteria, etc.), on the basis of which the matrix of judgments was compiled.

The aggregation of expert opinions, i.e., the generalization of the weights wi, taking into reason the opinions of all experts, can be performed, for example, using fuzzy weighted aggregation operators. The data in Table 5 seems to us to be subjective, therefore, requiring verification.

The following tables with rating matrices and indicator weights also require verification, because there is no clear and reliable research methodology in the work. It is not clear what the authors are analyzing: indicator weights? stage data? real costs at different stages of the construction life cycle? Table 9 "Complex weights for the index system for assessing low-cost construction for the reconstruction of industrial heritage" suggests that the authors analyze the weights of the indicators, and not the costs themselves.

Weight-based TOPSIS score needs to be normalized. It is not clear what happened to the data that they require normalization? We guess that the authors are faced with the creation of a multi-criteria decision making model with fuzzy parameters. However, the authors do not suspect that the quantitative indicators of input and output data have different dimensions, therefore, in order to compare alternatives, a normalization process is necessary, which will allow them to be brought to a comparable form.

The authors should develop a step-by-step TOPSIS estimation algorithm that allows to operate with ambiguous fuzzy input data, subjective expert opinions. The concept behind the TOPSIS method is that the best alternative should be the closest to an ideal solution and the furthest away from a negative ideal solution. Authors must provide a detailed description of the procedure that determines the difference between fuzzy sets; procedures for identifying anomalous values as errors of the 1st kind and their elimination using standard methods; a scheme for comparing data on the levels of negative impact on low-cost construction.

We understand that the condition for a comprehensive assessment, taking into account the weights, is their normalization. However, the authors must justify the use of 2 formulas in detail. We do not see the application of the necessary procedures in the case of normalized fuzzy criteria weights. It is necessary to define the ideal solution N+ and the negative ideal solution N-. In each case, a measure of the distance Nk of the complex estimate from the ideal solution, including the proximity factor, should be determined step by step.

The same work must be done with respect to the AHP method. A characteristic feature of the first is that it allows you to compare the selected criteria and build a comparison matrix. With its help, at the next stage, global and local preferences are determined and the matching factor is calculated. At the final stage of the assessment, a final rating of acceptable alternatives is compiled based on the calculation of their utility function. The calculations for the rating should be submitted step by step.

We should see a clear justification for using the AHP method, which often involves subjective determination of the weight of criteria depending on the opinions of experts, gives rise to problems in determining the interdependence of alternatives and criteria. All this can lead to non-compliance of the evaluated options with the ranking criteria and inversion of the rating.

Conclusions should be rewritten in the light of realistic and credible due diligence procedures.

The References need updating and selection of sources. For example, the source "9 Harfst, J.; Sandrister, J.; Fisher, W. Industrial tourism as a driver of sustainable development? ...” has only an indirect significance for the study, since the authors are not studying tourism, but assessments using the AHP and TOPSIS methods.

Author Response

(The authors gave the same response as above.)

Round 2

Reviewer 2 Report

Dear authors,
The paper has been revised and reworked. I believe it can be published in this form if the other reviewers agree.

Author Response

Dear Reviewer:

We appreciate your insightful comments From the beginning of the revision of the article and thank you for giving us the opportunity to revise it.

The specific reply will be sent with the attachment, please refer to it.

Thank you again for your efforts.

Sincerely,

Manuscript authors

Reviewer 4 Report

The authors have done a great work of improving the article. However, how to give the exact title of this article? Authors should think about this and structure the revised version of their paper accordingly. In our opinion, the name remains unclear. It could be called Multicriteria Model for Determining the Best and Inexpensive Industrial Transformation under Fuzzy Inputs, but it is up to the authors how to change the name.

Author Response

Dear Reviewer:

We appreciate your insightful comments From the beginning of the revision of the article and thank you for giving us the opportunity to revise it.

The specific reply will be sent with the attachment, please refer to it.

Thank you again for your suggestions and efforts to revise the article.

Sincerely,

Manuscript authors
